# Thiabendazole and Thiabendazole-Formic Acid Solvate: A Computational, Crystallographic, Spectroscopic and Thermal Study

**DOI:** 10.3390/molecules25133083

**Published:** 2020-07-06

**Authors:** Andreia M. Tabanez, Bernardo A. Nogueira, Alberto Milani, M. Ermelinda S. Eusébio, José A. Paixão, Hayrunnisa Nur Kabuk, Maria Jajuga, Gulce O. Ildiz, Rui Fausto

**Affiliations:** 1Department of Chemistry, CQC, University of Coimbra, P-3004-535 Coimbra, Portugal; andreia.tabanez@bluepharmagroup.com (A.M.T.); ban@qui.uc.pt (B.A.N.); quierme@ci.uc.pt (M.E.S.E.); jajuga.maria@onet.pl (M.J.); g.ogruc@iku.edu.tr (G.O.I.); 2Dipartimento di Chimica, Materiali e Ingegneria Chimica “G. Natta”, CMIC, Politecnico di Milano, 20133 Milano, Italy; alberto.milani@polimi.it; 3Department of Physics, CFisUC, University of Coimbra, P-3004-516 Coimbra, Portugal; jap@fis.uc.pt; 4Department of Physics, Faculty of Sciences and Letters, Istanbul Kultur University, 34158 Istanbul, Turkey; hayrunnisa1nur@gmail.com; 5Department of Chemistry, King Fahd University of Petroleum and Minerals, Dhahran 31261, Saudi Arabia

**Keywords:** thiabendazole, DFT calculations, formic acid, solvate, crystal structure

## Abstract

Thiabendazole (TBZ) is a substance which has been receiving multiple important applications in several domains, from medicine and pharmaceutical sciences, to agriculture and food industry. Here, a comprehensive multi-technique investigation on the molecular and crystal properties of TBZ is reported. In addition, a new solvate of the compound is described and characterized structurally, vibrationally and thermochemically for the first time. Density functional theory (DFT) calculations were used to investigate the conformational space of thiabendazole (TBZ), revealing the existence of two conformers, the most stable planar *trans* form and a double-degenerated-by-symmetry *gauche* form, which is ~30 kJ mol^−1^ higher in energy than the *trans* conformer. The intramolecular interactions playing the major roles in determining the structure of the TBZ molecule and its conformational preferences were characterized. The UV-visible and infrared spectra of the isolated molecule (most stable *trans* conformer) were also calculated, and their assignment undertaken. The information obtained for the isolated molecule provided a strong basis for the understanding of the intermolecular interactions and properties of the crystalline compound. In particular, the infrared spectrum for the isolated molecule was compared with that of crystalline TBZ and the differences between the two spectra were interpreted in terms of the major intermolecular interactions existing in the solid state. The analysis of the infrared spectral data was complemented with vibrational results of up-to-date fully-periodic DFT calculations and Raman spectroscopic studies. The thermal behavior of TBZ was also investigated using differential scanning calorimetry (DSC) and thermogravimetry. Furthermore, a new TBZ–formic acid solvate [2-(1,3-thiazol-4-yl)benzimidazolium formate formic acid solvate] was synthesized and its crystal structure determined by X-ray diffraction. The Hirshfeld method was used to explore the intermolecular interactions in the crystal of the new TBZ solvate, comparing them with those present in the neat TBZ crystal. Raman spectroscopy and DSC studies were also carried out on the solvate to further characterize this species and investigate its temperature-induced desolvation.

## 1. Introduction

Thiabendazole (TBZ; C_10_H_7_N_3_S; IUPAC name: 4-(1*H*-1,3-benzodiazol-2-yl)-1,3-thiazole; Scheme 1) is an organic chemical compound derived from benzimidazole and thiazole. It firstly gained commercial importance when Brown et al. reported that the compound, prepared by the reaction of 4-thiazolecarboxamide with *o*-phenylenediamine, using a polyphosphoric acid catalyst, exhibited broad spectrum anti-helminthic activity [1,2]. TBZ is rapidly absorbed upon ingestion, and the peak plasma concentration is reached within 1 to 2 hours after the oral administration of a suspension. Efficient absorption of the compound also takes place through topical preparations applied to the skin [3]. If the recommended dose is not exceeded, the compound has no harmful effects on the body, being hydrolyzed in the liver and excreted by the kidneys [3]. In an overdose, cells die by hepatocyte apoptosis, and this causes severe liver damage [3]. Symptoms of a thiabendazole overdose might include changes in vision and in behavior or personality [3]. TBZ is also known by the pharmaceutical brand names Mintezol, Tresaderm, and Acrobetec.

Besides its applications as a human or veterinary drug, the compound is also used as a food additive, to protect fruits against fungi (fungicide) and parasites [4,5,6,7]. TBZ was approved by FDA for oral use as an anti-fungal and anti-helminthic drug in 1967 [8].

TBZ presents structural features that also make it a potential candidate for other biological applications. Indeed, bearing three nitrogen atoms and one sulfur atom that may act as coordination centers, the compound is a good chelating agent, which means that it may also receive medicinal use in cases of metal poisoning, such as those caused by lead, mercury, or antimony, metals to whom TBZ binds easily [9]. Thiabendazole was also reported to block angiogenesis, and this property of the compound has been shown for some types of cancer [8,10].

In spite of its recognized practical interest, there are not many studies on the molecular properties of TBZ, and the physicochemical properties of the compound have also been only scarcely investigated. The crystal structure of thiabendazole was reported by Trus and Marsh [11], the crystal being orthorhombic, space group *Pbca*, with *a* = 17.052 (7), *b* = 10.998 (4) and *c* = 10.030 (8) Å, and 8 molecules per unit cell. Kim and co-workers [12] have reported both the Raman spectrum and the surface enhanced Raman spectrum (SERS) of the compound on silver, and investigated the influence of pH on the adsorption mechanism. They have shown that most of TBZ molecules were adsorbed on a silver surface by the π electrons in neutral and acidic conditions, but in acid conditions some molecules were adsorbed via the sulfur and nitrogen atoms tilted slightly to the surface. The terahertz spectrum of thiabendazole, at room temperature, has also been reported and assigned [13]. Santos Silva et al. [14] investigated TBZ (together with a few other compounds which exhibit anti-parasitic bioactivity) by infrared spectroscopy and thermogravimetry. The authors focused on analytical issues and investigated the stability of the studied compounds under different conditions. The infrared spectrum of TBZ was only described vaguely. The interaction of thiabendazole with montmorillonite was studied by Lombardi and co-workers [15], who also presented the powder X-ray diffraction pattern of TBZ and briefly discussed the differences between the infrared spectrum of montmorillonite intercalated TBZ and that of the sole compound. The last study deserving to be mentioned here is that of Wei et al. [16], where the syntheses, crystal structures, elemental and thermogravimetric analyses, and infrared spectra of four novel metal complexes with TBZ ligands were reported. In all the complexes investigated, thiabendazole acts as a bidentate chelate. To the best of our knowledge, no further studies have been reported hitherto focusing on TBZ structure-related problems.

In the present study, the conformational space of TBZ was characterized, and details of both the geometrical and electronic structures of the TBZ molecule were evaluated using density functional theory (DFT) calculations and spectroscopic methods. The room temperature crystalline phase of the compound was also characterized spectroscopically and its thermal behavior till fusion investigated. In addition, a TBZ-formic acid solvate of formula (TBZ-H)^+^.HCOO^−^.HCOOH [2-(1,3-thiazol-4-yl)benzimidazolium formate formic acid solvate] was synthesized and their crystal structure as well as spectroscopic properties and thermal behavior evaluated.

The formation of solvates is a common occurrence among organic compounds and has many practical implications, in particular for the pharmaceutical industry, since it affects the physicochemical properties of the materials, such as their density, melting point and dissolution rate, which can in turn influence its manufacturability and pharmacokinetic properties [17,18,19]. It has been estimated that around 33% of the known organic compounds have the ability to form hydrates, while about 10% are capable of forming solvates with organic solvents [18]. To the best of our knowledge, only the nitrate monohydrate solvate of TBZ has been described hitherto [20].

## 2. Materials and Methods

### 2.1. Experimental Details

Thiabendazole was acquired from Sigma-Aldrich (99% purity) and used without further purification. Formic acid, used in the preparation of the solvate, was purchased from Acros Organics (98% purity).

The preparation of the TBZ-formic acid solvate [2-(1,3-thiazol-4-yl)benzimidazolium formate formic acid solvate] was made by the dissolution of TBZ (30 mg) in formic acid (5 mL), followed by the evaporation of the solvent at room temperature, which lasted a few days, the obtained crystals being subsequently dried in a desiccator. The crystals were then examined using Raman microspectroscopy and shown to correspond to a mixture of two types of crystals, with clearly different morphologies. The crystals present in larger amount correspond to the original TBZ crystalline form, while those present in smaller quantity belong to the TBZ-formic acid solvate, as demonstrated by single crystal X-ray diffraction.

Differential scanning calorimetry (DSC) measurements were done using a Pyris-1 power compensation calorimeter from Perkin-Elmer, with an intra-cooler cooling unit at a −25 °C (ethylene glycol-water, 1:1 *v*/*v*, cooling mixture), under a 20 mL min^−1^ nitrogen purge flow. Open aluminum pans were used in this work (samples weight between 1.0 and 2.0 mg), and an empty pan was used as reference. Indium (PerkinElmer, 99.99%, *T*_fus_ = 156.60 °C) and biphenyl (CRM LGC, *T*_fus_ = 68.93 ± 0.03 °C) were used for temperature and enthalpy calibrations of the instrument. In these experiments, the samples were scanned from 25 to 315 °C at a scan rate of 10 °C min^−1^.

Attenuated total reflectance (ATR) infrared spectra (1 cm^−1^ resolution) were recorded using a Smart Orbit ATR accessory, in Thermo Nicolet 6700 Fourier transform infrared (FTIR) spectrometer, equipped with a Ge/KBr beam splitter and a deuterated triglycine sulphate (DTGS) detector. To avoid interferences from H_2_O and CO_2_, a flux of air free of water vapor and carbon dioxide continuously purged the optical path of the spectrometer.

Single crystal Raman spectra were obtained, in the Raman shift wavenumber range 50–4000 cm^−1^ (accuracy better than 0.5 cm^−1^) using a Raman micro-system Horiba LabRam HR Evolution. Excitation was provided by a HeNe laser (λ = 633 nm), the laser power at the sample being ~17 mW. The collection time was set to 30 seconds, with 30 accumulations being averaged to produce the final spectra. A 50× objective lens was used, giving a laser spot diameter of 1 μm at the sample. Calibration was done using the characteristic Si wafer band (520.5 cm^−1^).

The single crystal X-ray diffraction (XRD) measurements were carried out in a Bruker APEX II diffractometer, at 293(2) K, using graphite monochromated MoKα (*λ* = 0.71073 Å) radiation. Data integration and scaling were performed with the SAINT suite of programs, and SADABS was used for the data collection, which was based on the measurement of a large set of redundant reflections [21]. The structure was solved by direct methods using SHELXT-2014/5 [22], and full-matrix least-squares refinement of the structural model was performed using SHELXL 2018/1 [22]. All non-hydrogen atoms were refined anisotropically. Hydrogen atoms were placed at calculated idealized positions and refined as riding using SHELXL-2018/1 default values, except for those of the N−H and O−H groups that where fully refined isotropically. A summary of the data collection and refinement details is given in Appendix A. Crystallographic figures and tables (Appendix A) were produced using the Platon [23] or Mercury [24] programs. A CIF file containing the supplementary crystallographic data was deposited at the Cambridge Crystallographic Data Centre, with reference CCDC 1975920.

### 2.2. Computational Methods

All calculations performed on the isolated molecule of TBZ and TBZ-formic acid complex were performed using the Gaussian 09 program package [25], with the B3LYP functional (which includes the Becke’s gradient exchange correction and the Lee, Yang, and Parr correlation functional) and the 6-311++G(2p,2d) basis set [26,27,28]. The computed harmonic vibrational frequencies and intensities for these molecular systems were obtained at the same level of theory and scaled by the standard factor for this combination of method and basis set (0.978), to correct them mostly for the effects of basis set limitations and anharmonicity. Normal modes were approximately characterized by using the animation module of ChemCraft [29]. Natural bond orbital (NBO) analysis was done using NBO (version 3.1) [30,31], as implemented in Gaussian 09. Time-dependent DFT (TD-DFT) calculations [32,33] were used to compute the energies of the low-energy excited states of the TBZ molecule and predict its UV spectrum, and were done using the same method and basis set used in the performed structural and vibrational analyses.

Full geometry optimization of the crystal structures and the prediction of IR and Raman spectra of TBZ and TBZ-formic acid solvate crystals have been carried out using CRYSTAL17 [34,35] at the DFT/B3LYP level [26,27,28], with both the 6–31G(d,p) and pob-TZVP basis sets [27,36]. The empirical correction for dispersion interaction (DFT-D2) proposed by Grimme [37,38,39] was used in the calculations in order to consider van der Waals and other dispersion attractive interaction forces. The structures used as first guess for the calculations on the crystals were the one determined experimentally by Trus and Marsh, for TBZ [11], and that resulting from the XRD measurements reported in the present work for the TBZ-formic acid solvate. In all cases, normal frequencies calculation at Γ point have been done on the optimized geometries, as achieved by the diagonalization of the numerically calculated Hessian matrix. The DFT computed spectra were scaled using scaling factors of 0.972 and 0.949 (above and below 1800 cm^−1^, respectively), which were chosen by fitting the calculated frequencies of the most intense bands to those obtained experimentally. The predicted normal modes were included in the discussion presented below if the calculated intensity was >5 km mol^−1^ for the IR spectra and >5 Å^4^ amu^−1^ for the Raman spectra.

## 3. Results and Discussion

### 3.1. DFT Studies on the Isolated Molecule of Thiabendazole

#### 3.1.1. Structural Details

According to the B3LYP/6-311++G(2d,2p) calculations, TBZ exhibits two conformers, a planar *trans* form, and a non-planar *gauche* conformer that corresponds to two symmetry-equivalent minima (Figure 1). The optimized geometries of the TBZ conformers are presented in Table 1.

The major differences in both energies and geometries of the two TBZ conformers (see Table 1) result essentially from the existence in the higher-energy *gauche* conformer of a repulsive interaction between the closely located C2-H and N3-H groups (H^…^H distance, 2.485 Å), which in the *trans* conformer is replaced by the attractive N3-H^…^N2 interaction (see Figure 1). In geometric terms, this reflects first in the fact that the higher-energy conformer (*gauche*) is not planar, having the rings (benzoimidazole and thiazole) twisted in relation to each other by ±21.1°. In addition, these differences in the intramolecular interactions in the inter-rings region of the molecule lead also to larger angles associated with the H-C2-C3-C4-N3-H *pseudo*-ring in the *gauche* conformer (and smaller angles in the opposite side of the C3-C4 inter-rings bond), as well as in the longer C3-C4 bond length in this conformer.

The consequences of the different types of intramolecular interactions in the inter-rings region can also be noticed by comparing the charges (natural charges) calculated for the two conformers (Figure 2). While in the regions of the molecule far from the inter-rings fragment the atomic charges are similar in the two conformers, the charges on H3 and H2, and also on N1, N2, N3, and C2, clearly reveal the existence of repulsive interactions in the *gauche* conformer and attractive ones in the *trans* conformer in the inter-rings region. In the higher-energy *gauche* form, the charges of the two hydrogen atoms are less positive, while those of the N3 and C2 atoms are more negative, because the H3^…^H2 repulsion led to electron charge migration from these hydrogen atoms to the atoms to which they are bound. Additionally, the charges in N2 and N1 are less negative in the *gauche* form, due to the electrostatic repulsion between these two negatively charged atoms in this conformer (such interaction is absent in the *trans* form; see Figure 2).

In summary, one can say that, while in the *trans* conformer there are two major attractive interactions (of electrostatic nature) in the inter-rings region of the molecule, in the *gauche* conformer there are two repulsive ones: an essentially electrostatic-in-nature N1^…^N2 interaction and a stronger H2^…^H3 repulsive interaction that is both electrostatic and steric in nature. These interactions justify the considerably higher energy of the *gauche* conformer (see below), the above-mentioned geometric differences between the two conformers, and also the differences in the calculated charges on the atoms around the inter-rings bond in the two forms.

Figure 3 depicts the calculated potential energy profile associated with the internal rotation about the C3-C4 inter-rings bond. The *gauche* conformer is higher in energy than the *trans* conformer by 30.64 kJ mol^−1^. This energy difference reduces to 29.24 kJ mol^−1^ when the zero-point correction is taken into account, while the Gibbs energy at 1 atm and 298.15 K (room temperature) is 27.76 kJ mol^−1^. The energy barrier between the two degenerated-by-symmetry *gauche* minima is only 0.45 kJ mol^−1^, while the barrier separating the *trans* conformer form the *gauche* forms amounts to 38.20 kJ mol^−1^. The corresponding transition states are the planar *cis* conformation and the symmetry-equivalent structures with the N1-C4-C3-N2 dihedral angle equal to ±80.6°. It is important to note that the zero point vibrational energy associated with the torsional vibration about the C3-C4 inter-rings bond is 0.21 kJ mol^−1^ (17.67 cm^−1^), i.e., smaller than the height of the barrier separating the two symmetry-equivalent *gauche* minima, thus implying that the *gauche* structures correspond to physically observable conformers, though the torsional potential is rather shallow in the region around the two equivalent minima and the transition state (*cis*) separating them. In consonance with this result, and the increased torsional flexibility in the *gauche* conformer relatively to the *trans* conformer, the calculated entropy for the first conformer is higher than that of the most stable conformer (ΔS°_(*gauche*-_*_trans_*_)_ = 6.52 J K^−1^ mol^−1^).

Taking into account the calculated relative Gibbs energy of the conformers the expected *trans*:*gauche* population of the two conformers in the gaseous phase, at room temperature is 99.997:0.003, i.e., it can be assumed that only the most stable *trans* conformer is of practical significance.

It is also worth mentioning that the *trans* conformer is also the form that was found to be present in the room temperature crystalline form of the studied compound (see Figure 4 for a representation of the crystalline unit cell, as reported by Trus and Marsh [11]). Because of this, in this article only the properties of the *trans* conformer will be described from now on.

We have also performed calculations on the protonated form of thiabendazole, since as shown below, this is the species present in the newly synthesized thiabendazole-formic acid solvate. The cation was found to be planar, with a stabilizing N–H^…^N intramolecular hydrogen bond and a repulsive C–H^…^H–N interaction in the inter-rings region of the molecule. The energy of the cation was found to be higher by that of TBZ + H atom by 311.4 kJ mol^−1^.

#### 3.1.2. Spectroscopic Properties

The infrared spectrum of the isolated molecule of thiabendazole (*trans* conformer) is shown in Figure 5. Assignments are provided in Table 2. The proposed assignments were based on the analysis of the normal modes performed using the animation module of ChemCraft [29].

Some relevant characteristic intense bands deserving here a special mention are those calculated at: *(i)* 3573 (νNH), 1392, and 1167 (both bands with substantial contribution from the δN-H coordinate), and 509 (γN-H) cm^−1^, all these modes being associated with the N-H imidazole fragment, *(ii)* 1261 cm^−1^ (νC6-N1/νC5-N3), also associated with the imidazole ring, and *(iii)* 1422 (νC1-N2), 1305 (νC3-N2), 857 (νC2-S) and 820 (γC2H/γC1H symmetric mode) cm^−1^, all originated in the thiazole ring. The stretching vibration of the inter-rings C3-C4 bond was predicted at 1560 cm^−1^, while the calculated wavenumber for the torsion about this bond is 57 cm^−1^ and corresponds to the lowest vibrational frequency of the molecule (in agreement with the flexibility of the TBZ molecule about the inter-rings C3-C4 bond already mentioned above).

The UV spectrum of TBZ was also calculated in the present study, using the TD-DFT/B3LYP/6-311++G(2d,2p) method (Figure 6). Table 3 summarizes these calculations, including also data for spin-forbidden transitions to low energy triplet states.

The experimental UV spectrum of thiabendazole in different solutions has been determined experimentally by several authors, e.g., in methanol [40] and in chloroform [41]. The maximum of absorption in these two solvents was observed at 298 and 302 nm, respectively. The UV spectrum calculated in this work shows the intense HOMO→LUMO transition with a maximum of 307.01 nm, which fits quite well the experimental values [40,41].

The orbitals involved in the transitions reported in Table 3 are depicted in Figure 7, after localization using the Natural Bond Orbitals (NBO) approach. The HOMO corresponds to the N2 (azothiazole) lone electron pair orbital, while the LUMO is an anti-bonding type orbital essentially localized in the C4-N1 bond, so that the HOMO→LUMO transition implies some charge transfer from the thiazole ring to the benzimidazole fragment. The LUMO+1, on the other hand, is an anti-bonding orbital localized on the C5-C6 bond that belongs to both benzo and imidazole rings, so that the HOMO→LUMO+1 intense transition predicted at higher energy (271.17 nm) does also involve charge transfer from the thiazole ring to the benzimidazole fragment.

### 3.2. DFT Studies on the Isolated Thiabendazole-Formic Acid Complex

The studies performed in this work on the newly synthesized TBZ-formic acid solvate focused on the properties of the crystalline material. Nevertheless, DFT calculations were also performed on the isolated structural unit of the crystal of the solvate. As starting structure for the DFT calculations, atomic coordinates were extracted from the crystal structure obtained by in single crystal X-ray diffraction experiments described in detail below (Section 3.3 and Section 3.4).

The structural unit of the crystal of the solvate is formed by a protonated TBZ molecule (thiabendazolium cation), a formate anion, and a neutral formic acid molecule (see Figure 8). The thiabendazolium cation is hydrogen bonded to the formate anion through an NH^…^O bond, the oxygen atom of the formate ion involved in this hydrogen bond participating also as acceptor atom in one additional non-classic hydrogen bond of C-H^…^O type with the C2-H moiety of the thiazole ring of the thiabendazolium cation (H-bond distance, H^…^O = 2.510 Å), and being also involved in a short contact with the C-H fragment of the neutral formic acid molecule (H^…^O = 2.730 Å). The second oxygen atom of the formate ion acts as proton acceptor in an additional hydrogen bond in which the carboxylic group of the neutral formic acid molecule is the donor group (OH^…^O; H^…^O = 1.69(4) Å). As a whole, taking into account the above mentioned hydrogen bonds and also the indicated short contact, the structure comprehends two 7-atoms *pseudo*-rings sharing one of the oxygen atoms of the formate ion (see Figure 8). When this structure was submitted to optimization, it was found to relax to a complex which differs from the original one in two main features: (i) the NH^…^O bond between the thiabendazolium cation and the formate anion observed in the crystal structure becomes stronger in the optimized isolated complex, the N^…^O and H^…^O distances reducing from 2.620(2) to 2.536 Å and from 1.76(2) to 1.462 Å, respectively, while the N-H bond increases from 0.87(2) to 1.089 Å; (ii) the neutral formic acid molecule rotates so that while keeping the original OH^…^O bond (which becomes stronger; H^…^O = 1.576 Å) it establishes an additional non-classic hydrogen bond of C-H^…^O type with the C13-H15 moiety of the thiazole ring of the TBZ molecule (H^…^O = 2.366 Å). The optimized isolated complex has then one 7-atoms *pseudo*-ring and one 6-atoms *pseudo*-ring which share two atoms (one oxygen atom from the formate ion and H2).

The fact that the structural unit existing in the crystal of the solvate and the optimized isolated complex are fundamentally different shows that the species present in the crystal exists only under the stabilization effects acting in the solid state. The orientation of the neutral molecule in the two structures is very much illustrative of the relevance of packing forces, since in the unit of the crystal the molecule is forced to be oriented in such a way that the C-H bond faces the C2-H bond of the thiazole ring of the TBZ molecule. Such orientation, leading to close proximity of the two hydrogen atoms is repulsive in nature and must be superseded by stabilization due to packing in the crystal. The fact that the NH^…^O and OH^…^O hydrogen bonds (which are the stronger H-bond interactions in the structure) are weaker in the crystal than for the isolated complex is also an indication of the relevance of packing forces in the crystal. Indeed, this result indicates that the *per se* strongest stabilizing specific NH^…^O and OH^…^O interactions are sacrificed in some extent in favor of a more efficient global network of weaker favorable interactions. As it will be shown in the next sections, these interactions are mostly of dispersive type (H^…^H, C^…^H, and π–π staking interactions) and non-classic CH^…^O hydrogen-bonds.

### 3.3. Single Crystal X-Ray Diffraction Studies on TBZ-Formic Acid Solvate, Theoretical Predictions, and Comparison with the Crystal of Pure TBZ

The newly synthesized 2-(1,3-thiazol-4-yl)benzimidazolium formate formic acid solvate crystallizes in the monoclinic *P*2_1_/*c* space group, with one protonated TBZ molecule, one formate anion and one neutral formic acid molecule in the asymmetric unit cell, with *a* = 3.83390 (10), *b* = 22.1950 (6) and *c* = 15.3695 (4) Å (Figure 9). Each TBZ cation assumes a conformation similar to the minimum energy form for the isolated molecule, being essentially planar (inter-ring angle: 4.0 (3) Å), and exhibits two NH^…^O hydrogen bonds with two neighbor formate anions (one of these also involved in one weak non-conventional CH^…^O hydrogen-bond), and two weak non-conventional CH^…^O hydrogen-bonds with neighbor neutral formic acid molecules (one via the benzimidazole ring and the other via the thiazole ring), the bare O atom of the formic acid molecule acting as the H acceptor for these two bonds. The main hydrogen-bonding has already been described above and it repeats forming chains propagating along the *[2 0 –1]* axis. In addition, each formate anion is connected with the neutral formic acid molecule through a strong OH^…^O bond, thus acting as links between the chains forming layers as depicted in Figure 9.

The observed parallel stacking of the molecular layers is favored by strong π–π interactions between the electron clouds of the aromatic rings. The distance between the center of gravity of homologous rings projected in a direction perpendicular to the ring planes is 3.5 Å, the direct distance between such gravity centers being 3.8 Å, corresponding to a ring slippage distance comprised between 1.5 and 1.7 Å, values that are typical for moderately strong π–π interactions between such type of aromatic rings. The hydrogen bonding pattern is quite different to that observed in pure TBZ, where the only relevant hydrogen bond interaction is that mediated by the N-H group, the bare imidazole N atom of a neighbor molecule acting as hydrogen bond acceptor. Also, in pure TBZ the molecular packing is not layered, the hydrogen bonding linking molecules in columns with alternate orientations in a crisscross pattern and where adjacent columns pack loosely. A common feature between the two crystal structures is the N atom of the thiazolyl ring not participating in intermolecular interactions.

Since we were interested to calculate the infrared and Raman spectra of the crystals of TBZ and TBZ-formic acid solvate using periodic quantum chemical calculations in order to help interpretation of the experimental spectra (Section 3.6) the XRD determined crystal structure for each material was optimized at the DFT/B3LYP level using two different basis set (pob-TZVP and 6-31G(d,p)), with and without the D2 Grimme correction [37,38,39]. The Grimme correction was used in order to evaluate the effect of including dispersion forces into the description of the systems, and the different basis chosen to check the effect of basis set on the obtained results. As a simple test to assess the reliability of the theoretical models, the accuracy in the prediction of the unit cell parameters of the crystals can be evaluated. The results are presented in Table 4, and allowed to conclude that calculations performed at the B3LYP-D/6-31G(d,p) level produce the best results, justifying the choice of this theoretical model for the vibrational spectra calculations described in Section 3.6.

### 3.4. Hirshfeld Analysis of Crystalline TBZ and TBZ-Formic Acid Solvate

Hirshfeld surface based techniques, developed by Spackman and his co-workers, represent an innovative method to shed light on intermolecular interactions and to gain insight into crystal packing [42,43]. In the present study, the Hirshfeld surface analysis was carried out for both TBZ and TBZ-formic acid solvate crystalline structures. The Hirshfeld surfaces and their 2D fingerprint plots were generated using the CrystalExplorer 17.5 software [44], with the structure input files obtained in the CIF format.

Hirshfeld surfaces are obtained from electron distributions that are calculated as sums of spherical atomic electron densities [42,43,44]. In brief, the Hirshfeld surface of a molecule in a crystal defines the region where the electron distribution given by the sum of the electron densities of the spherical atoms of a given molecule (the *promolecule*) exceeds that from all other promolecules in the crystal. Structure-related properties can be mapped on the Hirshfeld surface. The normalized contact distance (*d_norm_*) is calculated from the distances of a given point of the surface to the nearest atom outside (*d_e_*) and inside (*d_i_*) of the surface (normalized by the corresponding van der Waals (*vdw*) radii), as defined by Equation 1, and allows the identification of the regions of the molecule where intermolecular interactions are more important [43,45]. Additionally, the combination of *d_e_* and *d_i_* in the form of a 2D-fingerprint plot allows to condense information about the intermolecular contacts present in the crystal [43,46,47,48]. The 2D-fingerprint plots provide a visual summary of the frequency of each combination of *d_e_* and *d_i_* across the surface of a molecule, thus indicating not just which intermolecular interactions are present, but also the relative area of the surface corresponding to each kind of interaction, which is a measure of the relative amount of each interaction in the crystal.
(1)$$d_{norm}=\frac{d_{i}-r_{i}^{vdw}}{r_{i}^{vdw}}+\frac{d_{e}-r_{e}^{vdw}}{r_{e}^{vdw}}$$

Figure 10, Figure 11 and Figure 12 present the Hirshfeld surfaces and the intermolecular contacts of the TBZ unit in both TBZ and TBZ-formic acid solvate crystals as given by the *d_norm_* values. The values of *d_norm_* vary from −0.7700 to 1.1280 a.u. in TBZ-formic acid solvate and from −0.5580 to 1.1414 a.u. in the neat TBZ crystal. The red region (where the distance between the atoms is shorter than the sum of their van der Waals radii) is observed for TBZ-formic acid solvate in the region corresponding to the strong NH^…^O hydrogen bond between the benzimidazole moiety of the TBZ cation and the formate anion, while in the TBZ crystal the red region is observed in the zone of the hydrogen bond established between the N-H group of the benzimidazole moiety of one TBZ molecule and the bare benzimidazole moiety nitrogen atom of an adjacent molecule in the crystal (see Figure 11).

The largest contribution to the overall crystal packing in both compounds results from H…H interactions (30.9% in TBZ-formic acid solvate and 32.3% in TBZ; see Figure 11 and Figure 12). The second most significant contribution is the H…O interactions for TBZ-formic acid solvate, 18.3%, and the H…C/C…H for TBZ (23.3%; this interaction corresponds to 16.5% for TBZ-formic acid solvate). H…N/N…H contacts correspond to 6.2% and to 21.4% of the total Hirshfeld surface of TBZ-formic acid solvate and of TBZ, respectively. These results indicate that, apart from H…H interactions, the N–H…O hydrogen bonding is the major intermolecular interaction in the TBZ-formic acid solvate, whereas in the TBZ crystal the major intermolecular contributions are the van der Waals interactions in general (H…H, H…C/C…H), ahead of the relatively weaker N–H…N hydrogen bond.

### 3.5. DSC Study of TBZ and TBZ-Formic Acid Solvate Crystalline Materials

According to the performed DSC studies, the melting of the pure thiabendazole occurs at *T*_onset_ = (302.4 ± 0.3) °C and, upon cooling, the recrystallization was observed at *T*_onset_ = (282.2 ± 0.8) °C (Figure 13). The results obtained from the thermal analysis of TBZ are then in agreement with those documented in the literature where values in the range 300–305 °C [average (302.0 ± 0.5) °C] have been reported for the melting point of the compound [40,49]. We have also observed that the compound partially sublimates during experiments, which has been confirmed by TG and powder XRD measurements (Appendix A).

The DSC analysis for the TBZ-formic acid solvate allowed to determine the desolvation temperature. The results are shown in Figure 14 and show that desolvation starts to occur at *T* ≈ 50 °C. After each DSC experiment, the sample kept within the used aluminum pan was analyzed by Raman spectroscopy, the obtained spectrum matching that of pure thiabendazole.

### 3.6. Infrared and Raman Spectra of TBZ and TBZ-Formic Acid Solvate Crystalline Materials

#### 3.6.1. Infrared Spectra

Figure 15 shows the room temperature infrared spectrum of crystalline TBZ. When compared with the computed spectrum of the isolated molecule this spectrum shows significant differences (compare Figure 5 and Figure 15), since in the crystalline material intermolecular interactions play an important role. As discussed in Section 3.3, the strongest intermolecular interactions are N3-H^…^N1′ hydrogen bonds (the apostrophe designates a neighbor molecule) where the N^…^N and H^…^N distances are 2.86 and 2.10 Å, respectively, and the N-H^…^N angle amounts to 156° [11]. These hydrogen bonds link molecules together in a crisscross fashion (see Figure 4) to form columns running along the *c* axis. Adjacent columns pack together relatively loosely; the shortest intermolecular distance between heavy atoms is a C1^…^N1′ contact of 3.39 Å between molecules related by a 2_1_ axis along the *b* axis.

It is important to note that in the crystal the dihedral angle between the two rings is 10°, so that the molecules are distorted when compared to the minimum energy structure for isolated molecule. The shortest contact distances between molecules belonging to different columns are S^…^C10′ (3.74 Å) and C2^…^C9′ (3.72 Å) [11]. As already pointed out, the nitrogen atom of the thiazole ring (N2) is not involved in a hydrogen bond. In fact, N1 is a better hydrogen bond acceptor than N2, since it tends to assume an electronic configuration approaching that of N3 (the other nitrogen atom of the imidazole moiety), in order to increase the symmetry and, hence, the resonance stabilization of the imidazole ring.

The most evident differences between the infrared spectra of the isolated molecule of thiabendazole and of its crystal are observed in the high-frequency region, where the spectrum of the isolated molecule has only three significantly intense bands, that are due to the NH stretching mode and the two higher-intensity phenyl ring CH stretching vibrations (see Figure 5 and Table 2), while the spectrum of the crystal exhibits a complex broad profile (extending from ca. 2250 to 3300 cm^−1^), which is characteristic of species where intermolecular hydrogen bonding and anharmonic couplings involving the higher-energy vibrations and lattice modes play major roles [50,51,52]. Nevertheless, the differences are also substantial when we compare the low-frequency region of the predicted spectrum for the isolated molecule and the experimental spectrum of the crystal.

In order to theoretically model the experimental spectrum, fully periodic boundary conditions DFT calculations were performed, as described in Section 2.2. The high-frequency region of the spectrum could not be reproduced accurately also by these calculations, mostly because the strong effects of anharmonicity (both electrical and mechanical). However, the calculations were able to reproduce very well the low-frequency region of the spectrum of crystalline TBZ (Figure 16). The agreement observed between the experimental and the predicted spectra is indeed very good both regarding frequencies and relative intensities, which demonstrates the reliability of the used computational approach (see also the proposed assignments, presented in Table 5).

The comparison of the low-frequency data for the TBZ crystal with the predictions for the isolated molecule (compare Table 5 and Table 2) allows to conclude that the larger frequency shifts occur for both the in-plane bending and out-of-plane rocking modes of the NH group, which shift from 1167 cm^−1^ and 509 cm^−1^ in the isolated molecule to 1403 cm^−1^ and 902 cm^−1^ (experimental values; calculated: 1422 and 946 cm^−1^), respectively. These results are in agreement with the involvement of the NH group in the strongest intermolecular interaction in the crystal (the N3-H^…^N1′ hydrogen bond). As usually [50,51,52], the frequencies of both deformational modes increase in presence of the H-bond interaction, with the frequency shift being considerably larger for the out-of-plane vibration (393 cm^−1^) than for the in-plane mode (236 cm^−1^).

The IR spectrum of the TBZ-formic acid solvate crystal was also obtained using the same computational approach as for that of the TBZ crystal and in shown in the Appendix A. Unfortunately, the experimental IR spectrum of the solvate could not be obtained, due to occurrence of pressure-induced fast desolvation both during preparation of a KBr pellet of the compound (for transmission mode spectra collection) and during the attempted ATR experiments. Therefore, the discussion of the vibrational data of this species will be focused on its Raman spectra.

#### 3.6.2. Raman Spectra

The experimental and predicted Raman spectra of crystalline TBZ and TBZ-formic acid solvate are shown in Figure 17 (in the 200–1800 cm^−1^ region, with the high-frequency region drawn as inset). Table 6 summarizes the spectral data and presents the proposed assignments. In the high-frequency region, for both TBZ and TBZ-formic acid solvate all intense bands due to the νCH stretching modes predicted by the calculations are observed experimentally, while those due to the νOH and νNH stretching vibrations are extensively broadened in the experimental spectra due to H-bonding and anharmonic effects so that they are difficult to discern from the baseline. The small bands observed in both experimental spectra at about 3150 cm^−1^ can be assigned to the 1st overtone of the phenyl ring νC=C stretching mode whose fundamental is observed at 1593 and 1640 cm^−1^ in TBZ and in TBZ-formic acid solvate, respectively. In the lower frequency region, it can be noticed that, for most of the vibrations common to both crystals, there is a small wavenumber shift in going from TBZ to the TBZ-formic acid solvate. The largest differences between are observed for the bending NH vibrations (both in-plane and out-of-plane modes), as expected taking into account that the NH moiety participates in intermolecular H-bonds of different strengths in the two crystals, and also the well-known sensitivity of these modes to intermolecular interactions [50,51,52]. In the spectrum of the solvate, the most characteristic bands due to the vibrational modes of the formic acid and formate anion are observed at 1732, 1452, 1375, 1225, 1048, 699, and 198 cm^−1^ and 1387, 1359, and 1052 cm^−1^, respectively (see Table 6 for detailed assignments of these bands). These bands can be used as marker bands for the comparison of the spectra of the two crystals.

As it is also observable in Figure 17 (and also in Table 6), the predicted Raman spectra for both TBZ and TBZ-formic acid solvate show a fairly good agreement with the experimental spectra, in particular in what concerns frequencies.

Temperature variation Raman spectroscopy studies were also performed for the TBZ-formic acid solvate. A crystal of the solvate was isolated and heated from 25 °C to 315 °C, at a rate of 10 °C min^−1^ between spectra collection, so following temperature ramps approaching the conditions used in the DSC experiments described in Section 3.5. Raman spectra were collected at each increment of 25 °C in the temperature, except for the final temperature. The main purpose of these experiments was to obtain additional data on the transformations taking place during the heating and re-cooling of the TBZ-formic acid solvate, particularly the desolvation process (which takes place upon warming).

The obtained results are summarized in Figure 18. From this figure, it is possible to conclude that desolvation starts to be spectroscopically detectable around 50 °C and are complete at about 100 °C, since the intense characteristic bands of solvate (1290, 1469, 1598, and 1640 cm^−1^) start to disappear at 50 °C and are no longer visible at 100 °C; the emerging bands correspond to those of pure TBZ. These results are consistent with those obtained from DSC, though in the experimental conditions of the Raman measurements desolvation was completed at a lower temperature. This apparent discrepancy results from the fact that in the DSC experiments heating was done at a constant heating rate, while in the temperature variation Raman analysis the sample was heating by steps, as mentioned above. Though the temperature ramps were done at the same heating rate as in the DSC experiment, after each 25 °C increment in temperature a Raman spectrum was recorded after temperature stabilization. In practice, in the Raman experiments the sample experiences a sequence of annealing steps, which can be expected to favor a faster desolvation, as observed experimentally.

The sample was heated until 315 °C (above the melting temperature of TBZ) and, then, cooled to determine the structure of TBZ after recrystallization. As it can be seen in Figure 18, the spectrum of the melted sample does not differ substantially from that of the crystalline net TBZ, though the bands show the usual broadening associated to a crystal-to-liquid-phase transformation. After re-cooling the sample to room temperature, the already characterized crystalline form of neat TBZ is formed, though the spectrum immediately obtained after the cooling clearly reveals that full relaxation of the crystal is not promptly attained. Spectra of the same sample collected after a few hours are virtually identical to that of the commercially obtained TBZ sample.

## 4. Conclusions

DFT/B3LYP/6-311++G(2d,2p) calculations performed on the isolated molecule of thiabendazole indicated that the molecule may exist in two distinct conformers: the lowest energy planar *trans* form, and a double-degenerated-by-symmetry non-planar *gauche* conformer ~30 kJ mol^−1^ higher in energy than the conformational ground state. The repulsive interactions between the atoms closely located to the C3-C4 inter-rings bond in the *gauche* conformer vs. the attractive ones in the *trans* conformer were found to be the intramolecular interactions playing the major roles in determining the differences in the geometries of the two thiabendazole conformers as well as their relative energy. The infrared spectrum of the *trans* conformer was also calculated and assigned, and it was compared with that of the crystalline thiabendazole at room temperature in order to access the major intermolecular interactions existing in the solid state. The UV spectrum of the molecule was also predicted and shown to be in good agreement with existing experimental data for the compound in solution. The most intense band in the spectrum could be assigned to the HOMO→LUMO transition, which was found to involve partial charge transfer from the thiazole ring of the molecule to the benzimidazole fragment.

DFT calculations were also undertaken for the isolated main structural unit of the crystal of the newly synthesized TBZ-formic acid solvate, and the results compared with the motif found in the crystal. The fact that the structural unit existing in the crystal of the solvate and the optimized isolated complex are fundamentally different stressed the relevance of the intermolecular interactions in the solid state. These interactions were evaluated in details by means of Hirshfeld analysis based on the XRD structural data obtained for the crystalline TBZ-formic acid solvate, which stressed the relevance of dispersive-type intermolecular interactions in addition to H-bonding. The Hirshfeld results for the crystal of TBZ also pointed out for the relevance of dispersive-type interaction in this crystal, which, however, has a substantially different H-bond network when compared to the TBZ-formic acid crystal, as revealed by the XRD structural data. A common feature between the two crystal structures is the N atom of the thiazolyl ring not participating in intermolecular interactions.

The two materials, TBZ and its formic acid solvate, were also evaluated by differential scanning calorimetry, in particular to evaluate the desolvation process in the latter. It was found that desolvation starts to be observed at *T* ≈ 50 °C, giving rise to the same crystalline form of TBZ as acquired commercially. These results were confirmed by temperature variation Raman studies performed on the solvate. The two compounds were also investigated by Raman spectroscopy. Interpretation of the obtained Raman spectra (and also of the IR spectrum of crystalline TBZ) was helped by periodic DFT-D2 calculations.

This is the first comprehensive investigation on the molecular and crystal properties of thiabendazole and of its newly synthesized formic acid solvate, and the obtained data will certainly constitute a major source of fundamental structural and spectroscopic information for those working on biological applications of TBZ, in particular for those interesting to address its mechanisms of action at a molecular level.

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
