# Peer review of "Thiabendazole and Thiabendazole-Formic Acid Solvate: A Computational, Crystallographic, Spectroscopic and Thermal Study"

_molecules, 2020, doi:10.3390/molecules25133083_

Round 1

Reviewer 1 Report

Tabanez et al in this manuscript titled Thiabendazole and Thiabendazole-Formic Acid Solvate: A Computational, Crystallographic, Spectroscopic, and Thermal Study have tried to elucidate the molecular and crystal properties of thiabendazole and of its newly synthesized formic acid solvate.

The study is systematically done and the data convincing. The authors focus is on the characterization of the conformational space of TBZ detailing both the geometrical and electronic structures and evaluating them. They accomplished these objectives by characterizing and investigated the room temperature crystalline phase of the compound spectroscopically and its thermal behavior till fusion. In addition, a TBZ-formic acid solvate of formula synthesized and evaluating their crystal structure, spectroscopic properties, and thermal behavior.

The observed Caveat:

 The authors fail to discuss how these structural modifications would enhance the anti nematode activity of  Thiabendazole and Thiabendazole-Formic Acid Solvate? it is well known that Thiabendazole is active against a variety of nematodes and is the drug of choice for strongyloidiasis. Unfortunately, the drug or molecule had drastic side effects in the CNS  and hepatotoxic potential. 

The authors will also have to emphasize the importance of these structural modifications in the conclusion.

Reviewer 2 Report

In this manuscript by Nogueira and co-workers, a comprehensive study (DFT, IR, Raman, DSC and single crystal X-ray diffraction investigations, as well as Hirshfeld analysis) on thiabendazole and thiabendazole-formic acid solvate is described.

The paper is technically well-executed and useful from theoretical and experimental chemistry points of view. The manuscript is well-written and demanding, the referee only suggests some minor modifications:

line 66: write “[11]” instead of “[11]”

line 68: write “[12]” instead of “[12]”

line 540: write “three” instead of “3”

In summary I recommend that the paper be accepted for publication in Molecules after suggested minor modifications.

Reviewer 3 Report

The manuscript describes structural characterization of thiabendazole, a heterocyclic compound with multiple biological activities. First, the structures of two conformers of isolated thiabendazole molecule optimized at the DFT level are discussed together with their calculated UV and IR spectra. Then the crystal structures of thiabendazole and its newly prepared formic acid solvate is thoroughly discussed. Experimental IR and Raman spectra are compared with those calculated with the help of periodic DFT calculations. The desolvation of the solvate is studied by DSC and variable-temperature Raman spectroscopy.

The manuscript brings a detailed structural characterization of an important molecule and its solid forms. It deserves publishing in Molecules after addressing a few minor issues listed below.

  • Unify the citations of references in the text. (Numbers in square brackets or without brackets. Citations before or after punctuation.)
  • Check all Greek symbols in the text; they are not shown correctly in the pdf obtained for review (e.g. Figure 4 caption, Table 2 body and footnotes).
  • Check Figure 7; it does not correspond to the discussion in lines 328–335. (Text: “LUMO is an anti-bonding type orbital essentially localized in the C4-N1 bond.” However, LUMO in Figure 7 is localized at C7–C8 bond. HOMO is localized in the azothiazole ring and LUMO+1 is localized in the benzoimidazole fragment, but the authors claim that a transition from HOMO to LUMO+1 does not involve a charge transfer.
  • The axes labelling in Figures 11 and 12 (de, di) is not well visible. What is the color coding in these two figures?
